# Physical Performance Tests in Adult Neck Pain Patients with and without Clinical Myelopathic Signs: A Matched Case-Control Study

**DOI:** 10.3390/ijerph191610331

**Published:** 2022-08-19

**Authors:** Mon Mon Hnin Lwin, Rungthip Puntumetakul, Surachai Sae-Jung, Weerasak Tapanya, Uraiwan Chatchawan, Thiwaphon Chatprem

**Affiliations:** 1Human Movement Sciences, School of Physical Therapy, Faculty of Associated Medical Sciences, Khon Kaen University, Khon Kaen 40002, Thailand; 2Research Center in Back, Neck, Other Joint Pain and Human Performance, Khon Kaen University, Khon Kaen 40002, Thailand; 3School of Physical Therapy, Faculty of Associated Medical Sciences, Khon Kaen University, Khon Kaen 40002, Thailand; 4Department of Orthopaedics, Faculty of Medicine, Khon Kaen University, Khon Kaen 40002, Thailand; 5Department of Physical Therapy, School of Allied Health Sciences, University of Phayao, Phayao 56000, Thailand

**Keywords:** neck pain, grip and release test, nine-hole peg test, ten seconds step test, foot-tapping test

## Abstract

Patients with neck pain may experience cervical myelopathy, this may be detected by clinical myelopathic signs, although they did not have any symptom of myelopathy, except having neck pain. Decreasing physical performance is one symptom of cervical myelopathy that can lead to reduced quality of life in the elderly, however, in adult neck pain with clinical myelopathic signs have not been evaluated. Therefore, this research aimed to compare physical performance in two groups of adult patients with neck pain: those with and without clinical myelopathic signs. A total of 52 participants, gender, age, and body mass index (BMI) matched were allocated into 2 groups of 26 subjects with neck pain, those with, and without, clinical myelopathic signs. The grip and release test, nine-hole peg test, ten second step test and foot-tapping test were evaluated. The group of neck pain participants with clinical myelopathic signs exhibited greater impairment in all the tests than the group without clinical myelopathic signs (*p* < 0.001). Effect sizes (Cohen’s d) were grip and release test: 2.031, nine-hole peg test: 1.143, ten second step test: 1.329, and foot-tapping test: 0.798. Neck pain participants with clinical myelopathic signs demonstrated reduced physical performance. Physical performance tests may need to assessed in adult patients with neck pain who had clinical myelopathic signs.

## 1. Introduction

Neck pain has many adverse effects on patients. Importantly, it may be associated with cervical spine myelopathy (CSM) [1], a neurological injury to the spinal cord that may be caused by static mechanisms, dynamic mechanisms [2,3], or other serious pathologies [4,5,6]. A previous study reports that a quarter of axial neck pain and/or cervical radiculopathy cases progressed to symptomatic cervical myelopathy within a few years [7].

Normally, clinical myelopathy is used to describe the presence of myelopathic signs. Common signs include the Hoffman sign, inverted supinator reflex, Babinski reflex, Trömner sign, and finger escape sign. Each of these clinical tests for myelopathic signs has been compared with magnetic resonance imaging (MRI) results and proven sensitive in detecting cervical myelopathy in the elderly [8,9,10,11,12,13,14]. Later studies using myelopathic signs categorized adults with neck pain into groups with myelopathic signs present and myelopathic signs absent [15,16]. They concluded that myelopathic signs were associated with adult neck pain, presumably due to either irritation of the spinal cord or temporary spinal cord ischemia [15,16]. Thus, in adult patients without myelopathic symptoms (e.g., gait dysfunction, loss of hand dexterity, and motor/sensory dysfunction), the presence of clinical myelopathic signs may be used as the initial evaluation of CSM [15]. This may be because temporary spinal cord ischemia [17], ligament edema, and prolonged stretching of the cord and dura can induce patients to show clinical myelopathic signs [18].

Myelopathic symptoms are subjectively screened using questions in the clinical setting, but some questions will remain unasked due to the wide variety of symptoms that can occur in cervical myelopathy. Chikuda et al. (2010) [10] found that clinical myelopathic signs correlated significantly with the severity of myelopathy, particularly with the severity of motor dysfunction in the lower extremities [7], a finding supported by Elnomany (2016) [12]. In these studies [10,12], the investigators assessed upper and lower limb motor function using the Japanese Orthopedic Association (JOA) questionnaire, a subjective examination tool, and found that myelopathic signs were correlated with lower limb motor dysfunction from myelopathy [10,11,12]. However, the elderly patients who presented signs and symptoms in the JOA questionnaire may have had progressive CSM [10,12,19].

In an early study, Ono and colleagues (1987) [20] found that patients with myelopathy who demonstrated positive (abnormal) upper limb reflex jerks also had upper limb motor dysfunction. More recently, Nagata et al. (2012) found that physical performance measures were distinctly associated with cervical cord compression in elderly patients. They suggest that physical performance measures may be useful indicators in diagnosing the early stages of cervical myelopathy in the elderly [21].

Decreased physical performance is symptomatic of cervical myelopathy [22,23], and various tests can be used as screening tools for it, including the grip-and-release test (G&R), nine-hole peg test, 10 s step test, and foot-tapping test (FTT) [24,25,26]. Cervical cord compression has been found to influence lower limb physical performance, particularly in elderly patients [27]. As part of the aging process, most elderly persons already have reduced physical performance, which is not the case in normal adults [27]. From the point of view of physiotherapy, adult patients with neck pain commonly show myelopathic signs that decrease physical performance, but, to date, there remains a lack of scientific understanding of the physical performance differences, if any, between adults with neck pain with and without clinical myelopathic signs. Moreover, previous studies evaluated physical performance in aging patients with advanced CSM as determined by MRI [21], and it has not been assessed in patients with neck pain who have only clinical myelopathy signs that can be used to infer the pre-symptomatic condition of cervical myelopathy [8].

Machino and colleagues (2017) state that age and sex differences should be considered when using both the G&R and the 10 s step test [28]. Nikolaidis and colleagues (2019) conclude that an increased body mass index (BMI) is related to decreased physical performance [29]. Thus, controlling for age, sex, and BMI may strengthen the findings of the current study.

To the authors’ knowledge, this is the first case-control study examining physical performance in an adult neck pain population with and without clinical myelopathic signs. The study examined physical performance in adult neck pain patients with and without clinical myelopathic signs by matching age, sex, and BMI. The specific physical performance tests that can help to identify patients with CSM may prove useful in the clinical setting.

## 2. Materials and Methods

### 2.1. Ethics Statement

The study design, a matched case-control study, was approved by the following ethics review boards: Khon Kaen University Ethics Committee (HE 612278, Khon Kaen, Thailand, 28 November 2018), the University of Medical Technology, Ethical Review Committee (Mandalay, Myanmar, 21 September 2018), and the University of Public Health (Yangon, Myanmar, 30 November 2018). The study was also approved for registration with the Thai Clinical Trial Registry (TCTR20190121003).

### 2.2. Study Population Recruitment

Between December 2018 and April 2020, 52 Myanmar adults with subclinical neck pain (SCNP) were recruited in Mandalay, Myanmar. SCNP is defined as a low-grade neck dysfunction in which individuals have recurrent flare-ups of pain in the posterior neck region from the superior nuchal line to the spine of the scapula but have not yet sought regular treatment [30,31,32,33]. The participants were recruited through posted advertisements (on notice boards and social media) and announcements and were screened by a physiatrist. Those eligible and willing to take part in the study provided written informed consent before participation. As there was no comparable previous study reporting on the mean change of physical performance, a pilot study (10 participants, 5 in each group) was undertaken for sample size calculation. The sample size was calculated for each physical performance across four tests, and the highest numbers were selected. The sample size was calculated by the $n/\mathrm{group}=\frac{2{\left(Z_{\frac{\alpha \;}{2}}+{\;Z}_{\beta }\right)}^{2\;}\sigma^{2}}{{\left(\mu_{1}-{\;\mu }_{2}\right)}^{2}}$ formula using the mean difference in the G&R test between the two groups (μ1 − μ2 = 3.9) and a pooled variance estimation (σ^2^ = 18.58) from the pilot study, with 90% power and a significance level of α = 0.05. At least 26 participants were required in each group.

The participants were eligible for inclusion if they were aged 20–40 years and had a pain duration of more than three months in the past year. The exclusion criteria included a positive Spurling test, history of previous cervical spine surgery, severe neck pain (≥7.5 cm measured by visual analog scale), concurrent suffering from other locomotor disorders, a history of brain trauma, comorbid neurological diseases (such as cerebral infarction or neuropathy), consumption of any sedative drug or alcohol within the past 48 h, pregnancy, and myelopathic symptoms (such as tingling, numbness, weakness, loss of balance, loss of bowel/bladder function, or difficulty walking).

### 2.3. Screening Procedure

Screening tests were used to determine eligibility to participate in the study, and demographic data were recorded through direct interviews. Participants with neck pain were categorized into two groups (those with and those without clinical myelopathic signs) through the performance of four tests for clinical myelopathy signs: (i) Hoffman sign, (ii) Trömner sign, (iii) inverted supinator reflex, and (iv) Babinski sign. The test procedure and diagnostic performance of each clinical myelopathic sign are shown in Appendix A [1,8,9,34,35]. Participants who had at least one positive sign from the four clinical myelopathic tests were determined as having clinical myelopathy. Fifty-two subjects were matched on gender, age (±5 years), and BMI (±2 kg/m^2^). The subjects were allocated into one of two groups of 26 (1:1) having SCNP with and without clinical myelopathic signs. The participants’ progress through the study is illustrated in a flow diagram (Figure 1).

### 2.4. Assessment Procedure

The principal investigator (researcher M.M.H.L), who had 15 years of experience in spinal physiotherapy, assessed patients for clinical myelopathic signs. The assistant investigator, who had five years of general physiotherapy experience, was blinded to her colleague’s assessment and assessed the physical performance of all the participants using two validated functional tests suitable for use in the clinic and selected for the upper and lower limbs. The physical performance tests were the (i) G&R test, (ii) nine-hole peg test, (iii) 10 s step test, and (iv) FTT, which were randomized for each participant. The procedure for each physical performance test follows.

#### 2.4.1. Grip-and-Release Test

In the rapid G&R test, the participants were asked to grip and release their fingers with their forearm in pronation and wrist in mild extension (Figure 2). The number of completed cycles of movement within 10 s was separately counted on each side [24,28,36].

#### 2.4.2. Nine-Hole Peg Test

This test was undertaken with the participant seated at a table facing the center of a tray with both a shallow dish and holed board on the tabletop (Figure 3). The tray was placed so that the shallow dish, which held nine pegs, was on the participant’s dominant hand side, while the peg holes were on the non-dominant side. In the task, the participants took pegs from the dish and placed them in the nine peg holes on the other side of the tray. Instructions and a brief demonstration were provided. The participants took a practice test before completing the assessed task, using their dominant hand first, followed by testing using their non-dominant hand. In the test, the participants took the pegs sequentially and placed them in the pegboard until all nine peg holes were full. Then, again with the dominant hand, the nine pegs were removed one by one from the pegboard and put back into the dish without a rest. The time taken to complete the test was recorded using a stopwatch from the moment the participant touched the first peg until the last peg hit the dish. The test was then repeated with the participant’s non-dominant hand and the pegboard rotated so that the dish was in front of the non-dominant hand. Again, the time was recorded. All the participants took part. If a participant dropped a peg or the test was otherwise interrupted, the evaluator signaled the participant to stop, and a new test began [26,37,38].

#### 2.4.3. Ten Second Step Test

While standing, the participants were asked to march in place, raising their knees to 90° without holding anything to maintain balance. The steps were taken at maximum speed over 10 s (Figure 4). Participants who were apt to fall or exhibited anxiety about the test were asked to do this near a handrail. The participants were watched by a supervisor who stood beside them throughout the test [24,28,39].

#### 2.4.4. Foot Tapping Test

The participant was seated comfortably on a chair with hips and knees in the 90° flexion position (Figure 5). One foot at a time, the participants were instructed to keep their heels on the floor and tap their forefoot up and down. The number of taps completed in 10 s for each foot was counted and recorded separately for both sides [40,41].

### 2.5. Inter-Rater and Intra-Rater Reliability

Inter-rater and intra-rater reliability testing for the 4 physical performance tests described above was performed on 10 participants who were not included in the sample. To evaluate inter-rater reliability, the standard practice for evaluating the tests was performed by the assistant investigator before the initiation of the study. This was valuable for ensuring that the assistant investigator, who had 5 years of clinical experience, could evaluate the physical performance comparably to the principal investigator (researcher M.M.H.L.), who had 15 years of clinical experience. The physical performance tests were randomized for each participant, and a five-minute break was taken between researchers. The researchers were blinded to one another’s results.

To evaluate intra-rater variability, 10 patients with neck pain were reassessed twice by the assistant researcher 1 day after the inter-rater reliability assessment. A five-minute break was taken between tests.

### 2.6. Statistical Analysis

STATA version 10.1 (StataCorp, College Station, TX, USA) was used to analyze all the data. The demographic characteristics among the two groups were calculated for age, weight, height, and BMI. Descriptive statistics were used to explain the baseline demographics and study findings. The reliability of the physical performance tests was assessed using the intra-class correlation coefficient (ICC) (model 3.1) for continuous variables. Between-group comparisons of all the data were analyzed using the independent *t*-test with a significance level of *p* < 0.05. Effect sizes were calculated by Cohen’s d.

## 3. Results

The demographic data and clinical characteristics of all the participants showed no significant differences (Table 1) between the two groups (*p* > 0.05). Inter-rater and intra-rater reliability testing for the four physical performance tests showed excellent reliability, with an ICC of 0.80–1.00 [42]. The details of the ICC are shown in Table 2.

All the participants were right-handed nonsmokers. The physical performance of participants with neck pain with and without clinical myelopathic signs was assessed using four tests. Table 3 presents the mean differences and *p*-values of the two groups on the measures of the G&R, nine-hole peg, 10-s step, and FTT. Our results show significant differences between the two groups in all physical performance tests (*p* < 0.01). The differences and effect sizes (Cohen’s d) are shown in Table 3.

## 4. Discussion

This study compared the physical performance test results of neck pain patients with and without clinical myelopathic signs and found that all physical performance test results were significantly different (*p* < 0.01) between the two groups.

Most clinicians screen patients with neck pain to eliminate cervical cord compression by first asking questions in the subjective examination about possible cervical myelopathic symptoms, such as an inability to distinguish coins when removing change from one’s pocket, difficulty in buttoning, clumsiness of hands, gait disturbance, and urinary dysfunction [43]. The responses may lead the clinician to subsequently test for clinical myelopathic signs in their objective examination before providing appropriate treatment. If patients have subjective myelopathic symptoms and myelopathic signs, physical therapists direct them to physicians for MRI to get an accurate diagnosis. However, all the patients in this study did not report myelopathic symptoms, which may be initially misdiagnosed. A high degree of suspicion is required to include clinical myelopathy tests [43], such as the reflex tests conducted in the current study.

The G&R test should be used to quantitatively discern motor disability of the upper extremities and observe signs of laterality, as the results differed significantly (*p* < 0.001) between the two groups. This test was an accurate quantitative scale for assessing upper limb myelopathic signs. Fewer than 20 times in the G&R test were considered to be pathologic without considering their age [15]. The mean G&R test result score in the clinical myelopathic group in the current study was 17.5 times, which is less than the normal cut-off value of 20 times [44]. Notably, the result of the current study is in accordance with the mean G&R score of 18.2 times reported by Date et al. (2021) for mild and moderate cervical myelopathic groups [45].

There was a significant difference in the nine-hole peg test results between the two groups (*p* < 0.001). Chikuda et al., (2010) and Elnomany (2016) found that pyramidal tract signs (clinical myelopathic signs) were not correlated with upper limb motor dysfunction in myelopathy in the elderly [10,12]. In their studies, they subjectively assessed the upper limb motor function using the JOA questionnaire. In our study, we used the nine-hole peg test for quantitative measurement. Previous studies revealed the mean duration of the nine-hole peg test to be 18.62 ± 2.30 s [37] and median duration to be 11.9 ± 0.9 s [46] in healthy participants aged 20–40 years old and ≤45 years old, respectively. These results are slightly lower than what was observed for the myelopathic group in the current study, which scored an average time of 19.95 ± 2.26 s in the same test. The longer duration indicates a reduction of upper limb motor function.

The G&R test and nine-hole peg test results showed significant differences between neck pain groups with and without clinical myelopathic signs, indicating both tests could be used to clinically to detect early upper limb signs of cervical myelopathy.

In the current study, differences in the 10 s step test were apparent between the groups (*p* < 0.001). Machino et al., (2019) conducted the 10 s step test for participants in the age range between 40 and 80 years old and reported the cut-off value for 40-year-olds to be equal to 19 times [44]. In our study, the mean value of the 10 s step test was 18.46 times, which is slightly lower than the cut-off value (19 times) reported by Machino et al. [44]. In 2020, Cheng et al. reported 10 s step test results for cervical myelopathic patients to be <10 times in participants aged 55.50 ± 9.63 years [47]. In comparison, the average 10 s step test results of 18.46 found in the current study was greater than those reported by Cheng et al. (<10 times) and Nakashima et al. (14.5 times) [47,48]. This may be due to (i) their studies including participants displaying myelopathic symptoms/signs, whereas participants in the current study only showed myelopathic signs, and (ii) the participants in the aforementioned studies were older than those in our study [47,48].

In the current study, the FTT was remarkably different between the two groups (*p* < 0.01). The FTT is carried out chiefly by the distal muscles, which are required to produce voluntary, individual, and skillful movements. These muscles are controlled by the lateral descending system, which contains the corticospinal (pyramidal) and rubrospinal tracts [49]. Slowness of foot tapping is reported to be a more useful sign than the Babinski sign in identifying upper motor neuron weakness [25], and a reduction of FTT frequency indicates decreasing lower limb motor function. Clinical myelopathic signs occur due to disorders in the pyramidal tract. Because of these factors, the FTT results differed significantly between the two groups. An average result on the FTT of less than 18 times may have a greater chance of appearing in cervical myelopathy if there are no other diseases that can impair lower limb motor function [40]. In the current study, the average means of the FTT results in both groups were higher than 20. In the study by Numasawa et al. (2011), the participants with cervical myelopathy had already received surgical treatment, indicating their condition was one of severe myelopathy [40]. For screening cervical myelopathy, a mean value of the FTT of less than 18 should be considered.

The current study’s results are consistent with the findings of previous studies [10,12], as not all participants with clinical myelopathic signs had lower limb myelopathic symptoms. However, on average, the 10 s step test and FTT results in the group with clinical myelopathic signs were significantly lower than in the group without signs. Thus, the 10 s step test and FTT may be the most preferred clinical tests to assess the lower limbs in adult neck pain patients with clinical myelopathic signs.

Dinkeloo and colleagues (2020) found that smoking reduced physical performance [50]. Our physical performance results were not confounded by smoking, as our participants were nonsmokers. Reduced physical performance is a useful new diagnostic approach for determining the early stages of cervical myelopathy. In the current study, we asked the patients about myelopathic symptoms, and all the participants in the group with clinical myelopathic signs had no myelopathic symptoms. However, the results of the G&R, nine-hole peg, 10-s step, and FTT were significantly different between the two groups. According to our study results, clinical myelopathic signs may occur in adult patients with neck pain accompanied by reduced physical performance. Therefore, physical performance tests should be part of routine clinical examination for adult neck pain participants with clinical myelopathic signs.

There are some limitations in the present study. First, an MRI was not available to confirm whether the participants in the clinical myelopathic group had cervical myelopathy or other serious pathologies that can affect the spinal cord. Second, the sample comprised more females than males. Future research should involve a longitudinal study to follow up as to whether the adult neck pain patients with clinical myelopathic signs have myelopathy symptoms.

## 5. Conclusions

Adult neck pain participants with clinical myelopathic signs had significantly reduced physical performance compared with participants without clinical myelopathic signs. Therefore, physical performance tests should be used as alternative tests to classify whether patients have or do not have clinical myelopathy.

## Figures and Tables

**Figure 1 ijerph-19-10331-f001:**
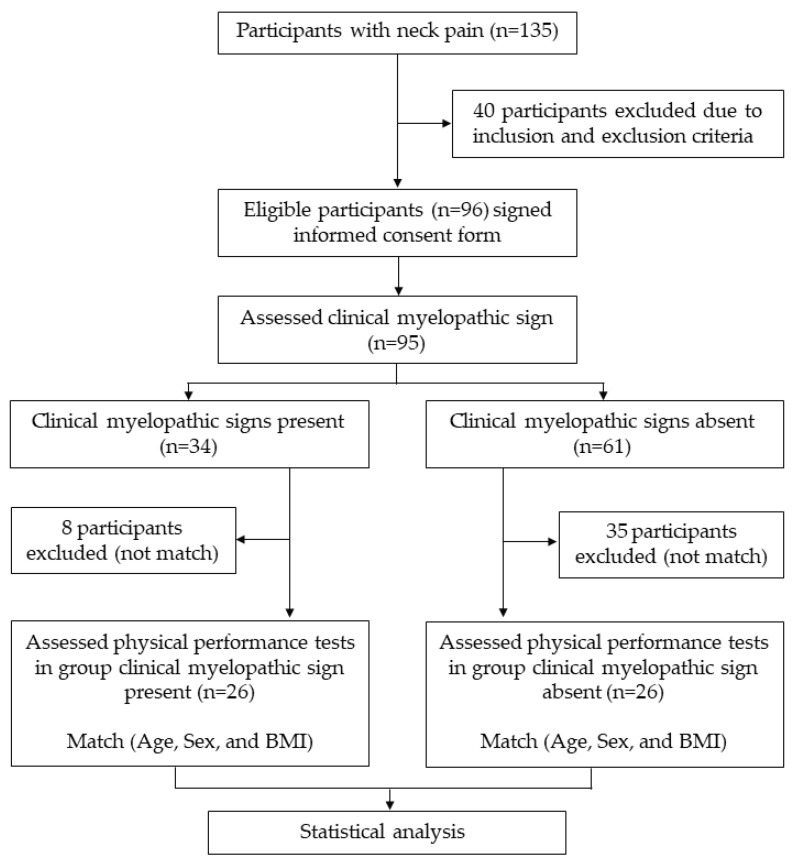
Overview of the study.

**Figure 2 ijerph-19-10331-f002:**
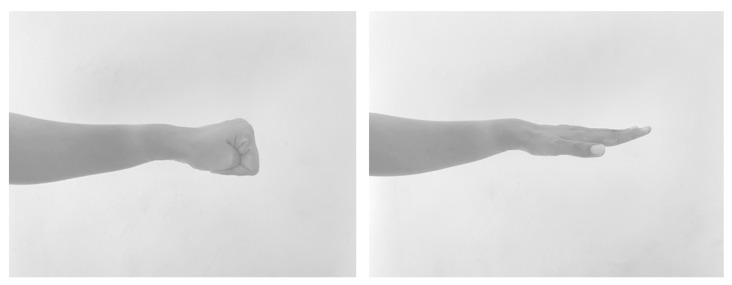
Grip and release test (G&R).

**Figure 3 ijerph-19-10331-f003:**
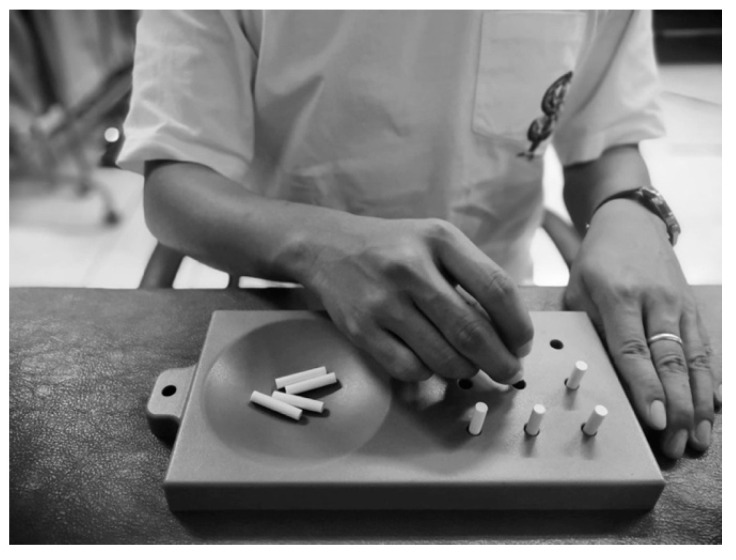
Nine-hole peg test.

**Figure 4 ijerph-19-10331-f004:**
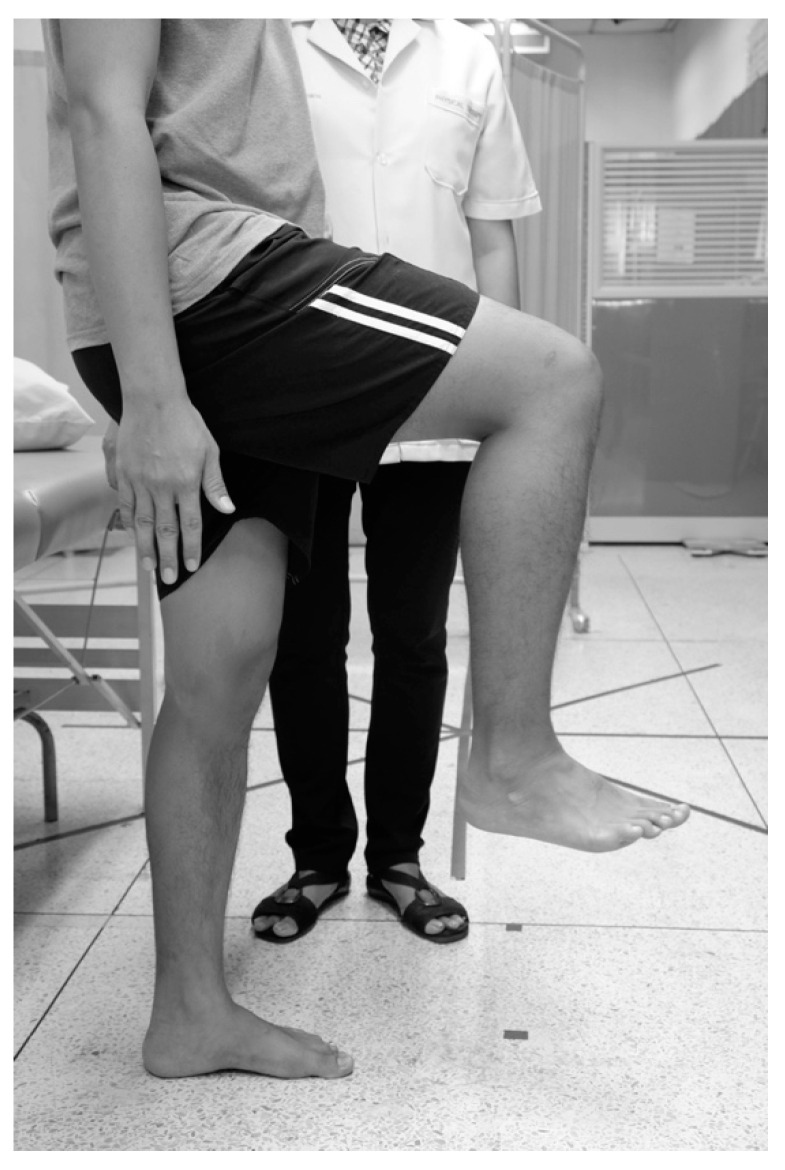
Ten Second Step Test.

**Figure 5 ijerph-19-10331-f005:**
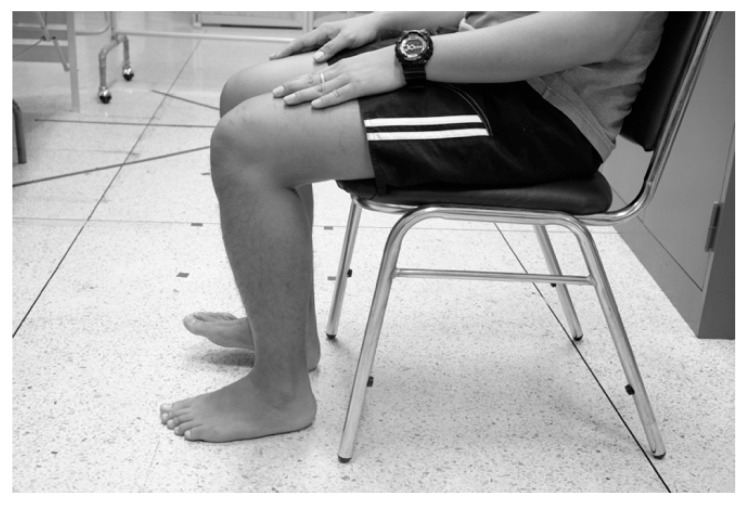
Foot Tapping Test.

**Table 1 ijerph-19-10331-t001:** General Characteristics of participants in neck pain groups with and without clinical myelopathic signs.

Variables	Neck Pain with Clinical Myelopathic Signs (*n* = 26)	Neck Pain without Clinical Myelopathic Signs (*n* = 26)	*p*-Value
**Age** (years), Mean ± SD	31.92 ± 5.25	31.81 ± 4.87	0.94
**Gender**, *n* (%)			
Male	7 (26.90)	7 (26.90)	
Female	19 (73.10)	19 (73.10)	
**Weight** (kg)	66.03 ± 10.62	66.66 ± 12.07	0.84
**Height** (cm)	159.97 ± 6.90	161.00 ± 6.19	0.58
**BMI** (kg/m^2^)	25.77 ± 3.67	25.61 ± 3.64	0.87
**Pain intensity**	4.91 ± 1.52	4.83 ± 1.70	0.86
VAS (cm)			
**Smoking**			
No	26 (100%)	26 (100%)	
Yes	-	-	
**Dominant Hand**			
Right	26 (100%)	26 (100%)	
Left	-	-	

**Abbreviations****:** SD, standard deviation; VAS, visual analog Scale; BMI, Body Mass Index.

**Table 2 ijerph-19-10331-t002:** Inter-rater and intra-rater reliability of physical performance test.

Test	Inter-Rater Reliability			Intra-Rater Reliability		
	ICC	95% CI	*p*-Value	ICC	95% CI	*p*-Value
**Grip and release right**	0.99	0.95 to 1.00	0.000	0.99	0.96 to 1.00	0.000
**Grip and release left**	0.97	0.88 to 0.99	0.000	0.97	0.88 to 0.99	0.000
**Nine** **-hole peg test right**	0.80	0.20 to 0.95	0.012	0.82	0.26 to 0.95	0.010
**Nine** **-hole peg test left**	0.95	0.79 to 0.99	0.000	0.95	0.78 to 0.99	0.000
**Ten seconds step test**	0.95	0.79 to 0.99	0.000	0.96	0.85 to 0.99	0.000
**Foot tapping test right**	0.96	0.84 to 0.99	0.000	0.93	0.71 to 0.98	0.000
**Foot tapping test left**	0.99	0.95 to 1.00	0.000	0.89	0.55 to 0.97	0.002

**Abbreviations****:** ICC, intraclass correlation coefficient; CI, confidence interval.

**Table 3 ijerph-19-10331-t003:** Mean differences of physical performance tests between groups with neck pain with and without clinical myelopathic signs.

Physical Performance Tests	Number	Mean ± SD	Mean Difference	95% CI	*p*-Value	Effect Size (Cohen’s d)
**Grip and Release Test** (times)						
CMG	26	17.50 ± 2.52	−4.88	−6.22 to −3.55	<0.001	2.03
NP	26	22.38 ± 2.28				
**Nine****-****Hole Peg Test** (seconds)						
CMG	26	19.95 ± 2.26	2.37	1.22 to 3.52	<0.001	1.14
NP	26	17.58 ± 1.87				
**Ten Seconds Step Test** (times)						
CMG	26	18.46 ± 1.79	−2.88	−4.10 to −1.67	<0.001	1.33
NP	26	21.35 ± 2.50				
**Foot Tapping Test** (times)						
CMG	26	23.70 ± 3.20	−2.70	−4.56 to −0.82	<0.01	0.80
NP	26	26.38 ± 3.51				

**Notes****:** Data presented as Mean ± SD, Mean difference, 95% CI, *p*-value from *t*-test, significant level was set as *p*-value < 0.05; **Abbreviations****:** CMG, Neck pain participants with clinical myelopathic signs; NP, Neck pain participants without clinical myelopathic sign.

## Data Availability

The data will be available for anyone who wishes to access them for research purpose and contract should be made via the corresponding author rungthip@kku.ac.th.

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
