# Peer review of "Physical Performance Tests in Adult Neck Pain Patients with and without Clinical Myelopathic Signs: A Matched Case-Control Study"

_ijerph, 2022, doi:10.3390/ijerph191610331_

Round 1

Reviewer 1 Report

This is an interesting and well-done study the authors made a great effort to evaluate each variable of this study. Despite the interesting study I have some questions that the authors need to resolve

First: Why authors didn't make any statistical study related to concordance between evaluators

Second: Why did the authors decide to include the variable smoking if they didn't have any patients with this condition, or smoking is a risk factor related to neck pain with myelopathic signs?

Third: The predominant female genre was an important finding, are the female genre is an important risk factor for neck pain in patients with myelopathic signs)

The discussion is well done and interesting authors discuss each point studied in their results and methodology, however, could be possible to discuss any relationship between myelopathic signs with neck pain in smoking patients, or why your think to not find any smoking patients in your study?

Author Response

Thank you so much for your feedback. These questions are valuable for us. The revised version of the manuscript is marked with yellow highlight changed for reviewer 1.

Reviewer 2 Report

The manuscript “Physical Performance Tests in Young Adult Neck Pain Patients with and without Clinical Myelopathic Signs: A Matched Case- Control Study” by Mon Mon Hnin Lwin et al. aimed to examine physical performances in young adult neck pain with and without clinical myelopathic signs by matching age, sex, and BMI.

Below are my comments and remarks regarding the manuscript:

1. Lack of information as to whether the patients had imaging tests, which is a condition for the diagnosis / exclusion of other causes of myelopathy

2. The severity of myelopathy was not taken into account, e.g. the number of symptoms / criteria: Hoffman sign, inverted supinator reflex, Babinski reflex, Trömner sign, and finger escape sign

3. Please provide a source which defined the criterion for the diagnosis of myelopathy on the basis of one of the given symptoms

4. Lack of text description of table 3; possibly move the full description from the discussion to the results section

5. Where is the answer to the  aim of the article - about age, sex, and BMI 

6. I would be cautious in describing conclusions in the absence of criteria for myelopathy.

Author Response

Thank you so much for your comments. We have carefully responded to answer all of your concerns. The revised version of the manuscript is marked with a grey highlight changed for reviewer 2.

Reviewer 3 Report

General Comments

The author did physical performance tests on myelopathic signs in young adult neck pain patients. However, the term "young adults" used in the research might mislead. According to the publication "Investing in the health and well-being of young adults", the age range for young adults is from 16-26 and/or 16-30 for a wider variety. Hence, the inclusion of 40-year-old patients means that they are out of the young adult age range.

Further reading

Investing in the health and well-being of young adults

https://www.ncbi.nlm.nih.gov/books/NBK284791

I found the research is proven helpful in determining the early myelopathy signs by conducting 4 physical performance tests: a grip and release test, a nine-hole peg test, ten seconds step test, and a foot-tapping test. Following up this research in detecting the myelopathic signs in young adult neck pain patients, it would be a good suggestion to further do an MRI check for cervical myelopathy confirmation. 

Specific comments:

Page 1, Line 22:

Taken from the abstract, there is a grammar mistake in this sentence, "Therefor this research aimed to compare physical performance in two...". Instead, it should be "Therefore."

Page 1, Line 33:

Since this research explains the physical performance test regarding the neck pain, I think "neck pain" should be added to the keyword.

Page 2, Line 78:

There is a grammar mistake in this sentence, "Form the..." It should be "From the point of view of physiotherapy..."

Page 3, Line 123:

In the sentence, "Participants were eligible for inclusion if they were aged between 20–40 years old." This shall follow up on the comment in the general comment section about the age range of young adults.

Page 3, Line 126:

The exclusion of, "severe neck pain" should be explained further as to distinguish it from the neck pain used for this study.

Page 7, Line 202:

Regarding inter-rater and intra-rater reliability tests, why are they performed only on 10 participants? Is this the sample size that takes the 5 participants from each group? If this is the sample size, the author should mention it before explaining further to avoid confusion about the number of participants.

Page 7, Line 213:

I think the author needs to specify what M.M.H.L means in this study to avoid confusion.

Page 8 Line 233 (Table 1) and Line 234

Is there any correlation between smoking habits and neck pain? Because table 1 mentioned the non-smoker participants in the variables, if there is a correlation, then the author has yet to mention the past studies concerning non-smoking patients.

Page 8, Line 235:

In the sentence, "One of the participants in the clinical myelopathic group reported symptoms (numbness) in their hands." This sentence does not meet the exclusion criteria for myelopathic symptoms listed on Page 3, Lines 124-129, “Participants were excluded if they had some conditions characterized by myelopathic symptoms, such as tingling, numbness, weakness...” Does the numbness occur before, during, or after the physical performance tests?

Page 9, Line 273-281:

The author has yet to explain the nine-hole peg test cut-off value. I think explaining is important so the readers can compare and understand the result better.

There are a few other minor grammatical errors. Please proofread and spell-check carefully.

Author Response

Many thanks for your feedback. We have carefully edited all your concerns. The revised version of the manuscript is almost marked with a green highlight changed for reviewer 3. However, grey (point 2) and yellow (point 10) highlights were used due to reviewer 2 and reviewer 1 comments.

Round 2

Reviewer 2 Report

I leave the decision to the editor

Author Response

Many thanks for your feedback

Reviewer 3 Report

The authors have provided a nicely detailed and thorough response to the comments from the previous review and have addressed my major concerns. However, the authors may consider comparing your results with other related papers in the discussion section.

Author Response

Thank you so much for your feedback and thank for the opportunity to give us edit our manuscript better. We have rewritten and added more previous data for comparison to our study as references number 46, 47, and 48. The revised version of the manuscript is almost marked with a yellow highlight change.
